# The Social Risk of High-Speed Rail Projects in China: A Bayesian Network Analysis

**Yutong Xue [1] and Pengcheng Xiang [1,2,3,*]**

[1]  School of Management Science & Real Estate, Chongqing University, Chongqing 400045, China; xueyutong_cn@163.com
[2]  International Research Center for Sustainable Built Environment, Chongqing University, Chongqing 400045, China
[3]  Construction Economics and Management Research Center, Chongqing University, Chongqing 400045, China
*  Correspondence: pcxiang@cqu.edu.cn; Tel.: +86-2365-120-848

**Abstract:** In China, high-speed rail projects have brought huge social and economic benefits to the affected regions after they are completed. However, the potential externalities of such projects cause competition for the station during the project planning phase, thus triggering social risks. This paper studies the mechanisms responsible for generating the social risk associated with such high-speed rail projects. Employing typical case studies, a social risk list for a given project is established. Based on the risk list, a Bayesian network model is developed and verified through case studies, expert interviews, and expert grading. Using the model's functions of reverse inference and sensitivity analysis, the key risk factors, sensitive risk factors, and maximum causal chain are identified. Countermeasures are then proposed to mitigate the social risk, such as increasing the transparency of and democratizing the planning process for high-speed rail projects, improving the mechanism by which local governments can express interest in such projects, and enhancing emergency management mechanisms. The findings provide points of reference for social risk management when it comes to planning high-speed rail projects and, more generally, offer significant guidance for socially sustainable decision-making processes for mega projects with massive externalities.

**Keywords:** high-speed rail project; social risk; risk analysis; Bayesian networks

## 1. Introduction

In the past two decades, world-renowned achievements have been made in the construction of China's high-speed rail (HSR) system. The development of HSR has, in turn, produced positive externalities. Regional connectivity has been improved, the concentration of economic activities and production factors in individual cities along the line has been accelerated, and economies of scale and scope have been formed [1]. According to a recent survey, compared to the situation prior to the construction of the HSR system, over 850 million new opportunities have been created to connect, trade, and exchange ideas each year, leading to additional economic activity, stronger innovation, and increased productivity [2]. The HSR system aims to enhance China's urbanization process by encouraging a complementary relationship among cities and allowing skills and technology to flow to smaller centers, thereby enhancing their competitiveness. By their transport connecting to national and international supply chains and innovation networks, a wide range of local stakeholders are benefited. For the general public, HSR means convenient transportation; for developers, HSR means higher housing prices; and for local governments, HSR represents political capital and local industrial development. Therefore, HSR is an immensely popular undertaking in such areas.

However, the massive externalities of HSR have caused social problems during the planning process. In the national Medium and Long-Term Railway Plan, the overall direction of the HSR (including starting and ending points for segments of the line) has been basically determined. What remains to be completed is clarification of the detailed distribution of HSR lines and stations. In the formal pre-feasibility study reports that are generated for this purpose, there are usually two or three different possible routes for the design of a local railway line and station through different regions, and one of them will be recommended by the experts. During this time, HSR planning is affected not only by objective factors such as geological and economic conditions, but also by the competitive game that plays out among stakeholders. Since 2009, because of the strong appeals from local populations and governments for the implementation of HSR transit in their areas, the "Battle for HSR" has not stopped. Thus, the HSR project, originally designed to be a public welfare project, has instead triggered several mass incidents, including demonstrations, petitions, traffic jams, and even conflicts between police and civilians. The Linshui 5/16 incident is a typical example. This incident occurred during the planning of the Dazhou-Chongqing HSR project in 2015. According to incomplete statistics, tens of thousands of people participated in the petition signing and subsequent march, 68 people (including members of the police force and civilians) were injured, and dozens of vehicles were burned [3]. In addition, there is evidence to suggest that new HSR lines are typically costly during the construction and operation stages [4]. Therefore, even though government plans to construct HSR lines can face public opposition, often organized through social actions, secondary risks to the projects such as locally incurred debts may also be realized during the later stages of these projects. To address such concerns, a ranking methodology to prioritize HSR corridors was proposed by Guirao and Campa and validated in Spain [5]. In China, however, there is no such approach to planning that considers social risks.

To date, research on HSR has primarily focused on the following four aspects [6]: (1) improvement of regional traffic accessibility [7,8]; (2) impact on the labor markets for commuting relations [9]; (3) promotion of industry and regional economy [10,11]; and (4) impact on the competitiveness of other transportation facilities [12]. The existing research has mainly focused on the social and economic effects after HSR is implemented. However, little attention has been given to the social risk bound up with the planning phase—risk generated, in turn, by the expected social and economic effects of HSR during its operational phase. Such risk needs to be studied and addressed. In the present study, the social risk of Chinese HSR projects is defined as the possibility of social instability associated with such projects, due to their potential social and economic externalities—and the active competition among stakeholders to which those externalities may lead during the project planning phase. To reduce the possibility for such social instability, the study analyzes the development process for HSR projects as well as the mechanisms responsible for the social risk they create. A Bayesian network (BN) model is developed to simulate the generation of social risk and the key elements of the risk-generation process are obtained. The findings are significant for governmental efforts to reduce social risk in HSR planning but, more generally, they offer important guidance for socially sustainable decision-making processes for mega projects that may cause social risk via the distribution of external benefits.

## 2. Literature Review

### 2.1. The Social Risk of Mega Projects

As the German social scientist Niklas Luhmann noted, we live in a society in which it is impossible not to take risks. As the frequency of human activities increases and the scope of those activities expands, the influence of our decision-making processes on the natural as well as social worlds has also been greatly enhanced [13]. The main components of the risk structure evolve from risk associated with nature to uncertainties arising from human endeavors [14]. Additionally, compared with general engineering projects, there are more stakeholders in complex mega projects. The stakeholders include not only the project developers and contractors but also local governments, community groups, and the public at large [15–17]. Thus, mega projects face many risks that may potentially carry a significant

impact, as well as a wide range of risk bearers. If these risks cannot be effectively controlled, they will spread to the larger society and threaten social stability [18]. Therefore, with respect to risk management for mega projects, attention should be given not only to risks related to cost overruns, time delays, safety incidents, and other such issues at the project performance level [19–21], but also to possible social risks. In previous studies, social risk has been defined as a possibility that can lead to social conflict, which, in turn, causes social instability and disorder [22]. If this possibility becomes concretized, social risk will be transformed into a social crisis, posing a threat to the sustainable development of human beings [23]. Further, Bohorquez defined social risk as the conditions or factors that generate vulnerability in communities in the wake of potentially destructive events [24]. In the Chinese context, Feng described social risk as the probability of political authority being threatened over a certain period [25]. In China, such social risk often manifests itself in the form of mass incidents, such as rallying, marching, petitioning, traffic blocking, collective fighting, and even more destructive collective violence, such as smashing, burning, or even murder.

Social risk management, defined as an institutional and social process that enables the convergence of policies, actors, strategies, and actions, reduces or ameliorates the conditions or factors that generate vulnerabilities in communities after potentially destructive events [16,24,26]. Current studies of social risk management focus on risk assessment [27–30]. By contrast, there are relatively few studies on the sources and mechanisms of social risk, although some previous work has explored the sources and mechanisms in question [31]. Regarding the sources of social risk, some scholars have argued that it derives from objective factors such as the technical problems of facilities and unscientific decision-making processes [32], whereas others trace the sources to subjective factors, such as the gap between people's desires and solutions for satisfying those desires [33], general psychological anxiety, or conflicts of interest among groups [34]. Regarding the generation or propagation of social risk, Roger Kasperson put forward the theory of social amplification of risk. He pointed out that the social expansion of risk includes two mechanisms: the mechanism responsible for the propagation of information about risks and the social-response mechanism involved in the uptake of that information [18]. Qu expounded the risk propagation or conduction mechanism systematically, and considered that the risk source, risk characteristics, risk carrier, and risk conduction path are the essential factors bearing on risk conduction [35].

Despite these contributions to the literature, however, key questions remain, especially for mega projects. What is the generation logic by which social risk is propagated from the stage of planning and location to the stage of public crisis? What are the key factors that affect social risk generation? These problems need to be resolved in engineering practice.

## 2.2. Modeling the Risk-Generation Mechanism

From the view of connection and development afforded by materialist dialectics, we can see that risks are not isolated and stationary but interconnected and dynamic [21]. The generation of risk is a process of qualitative change caused by quantitative change; this process is enabled by a mechanism that has been characterized in the domino theory as well as the energy transfer theory. Heinrich's domino theory states that accidents result from a chain of sequential issues [36], such that the process by which risk is generated can be described as a chain reaction [37]. Meanwhile, according to the energy release theory proposed by Hadden, the essence of risk generation is the process of energy transfer [38]. Once the energy exceeds the range that the system can bear, it will cause a risk explosion [39]. If we combine these two theories, we can describe the generation of risk as a sequential process, or chain reaction, along the following lines.

In a microscopic view, after a long period of accumulation, the energy associated with social risk bursts out when it breaks through the system's limit and then transfers to the next link, continues to accumulate once more, then bursts out and transfers again, and so on into wider and wider social spheres. In a more macroscopic view, this process can be divided into three parts: risk factors, risk events, and risk consequences [40]. Risk factors are the source from which risk events develop and a

mega project is a system with various risk factors in its internal and external environment [21]. If risk factors are not controlled, energy will accumulate and break through the system's "load-bearing" range, thereby being transmitted from risk factors to risk events [41]. The concept of risk events thus refers to the accidents or actions through which the energy of social risk is released into the wider socio-physical environment, leading to risk consequences. It is the middle link in the chain of risk generation. Risk consequence corresponds to the final loss of the project. Such loss is usually reflected in the deviation of the actual situation of the project from the expected goal [42]. Hence, cost overruns, time delays, and negative social impacts are all manifestations of risk loss. This macroscopic, three-part model of the mechanism responsible for risk generation is shown in Figure 1.

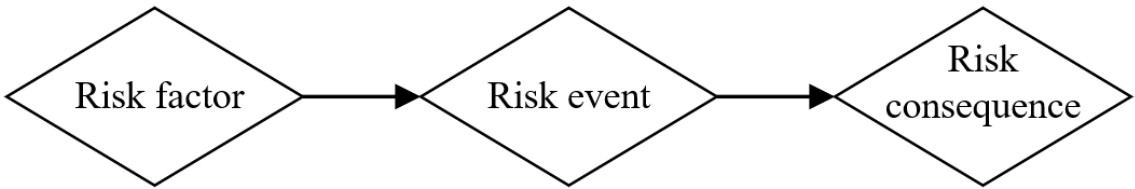

**Figure 1.** General mechanism of risk generation.

In terms of research methods that have been used to study the generation mechanism of social risk, qualitative methods are the most used [43]. Zhang proposed a social-amplification framework for social risks connected to engineering projects carrying high environmental risks; this framework includes the transmission, reception, and evolution of risk signals [44]. Based on the theory of social combustion, Xiang constructed a social risk evolution model during the phase of project construction, including the combustion substance, combustion aid, and ignition temperature that cause social risks [45]. Meanwhile, drawing on psychological studies of anxiety, Tan analyzed the mechanism responsible for the social risk of NIMBY (not in my back yard) projects, from breeding, through spreading, to diffusion [46]. These studies outline productive theoretical frameworks for the study of social risk generation and its significance. However, as noted previously, during social risk generation, there are complicated relationships among risk factors, risk events, and risk consequences, and multiple stakeholders are involved. Thus, working only with qualitative models makes it difficult to develop a complete explanation of the risk-generation process. In the present study, a BN analysis is used to overcome this limitation, by explaining the process of social risk generation via the concept of a risk network, in terms of which the mechanism involved can be more fully described and explained.

## 3. Methodology

The Bayesian network (BN), also known as the belief network, is a method based on probabilistic statistics that is used for analyzing data associated with complex uncertainty problems. A BN is a directed acyclic graph composed of nodes and directed edges connecting these nodes [47]. Variables are represented as nodes and the causal relationship between nodes is represented as directed edges, which generally point from a parent node to a child node. A BN consists of two parts: first, a qualitative description of the problem to be solved (referred to as the BN structure) and, second, a quantitative calculation designed to solve the problem (referred to as the network parameters). The reliability of BNs has been verified through many engineering applications, such as schedule risk management [48], project portfolio risk analysis [49], and stakeholder impact evaluation [50]. Generally speaking, BNs have the following advantages [48]: (1) small samples, as well as incomplete and noisy data sets, can be handled; (2) can be developed using expert opinions instead of historical data and still make reliable predictions; (3) can readily calculate the probability of events before and after the introduction of evidence and update, as necessary, their diagnosis or prediction; and (4) graphs are used to describe the interrelationships among nodes, making those interrelationships intuitive and easy to understand. Therefore, BNs provide an appropriate tool for studying the mechanism that generates social risk in the context of HSR projects.

The present study aimed to analyze the generation mechanism of social risk for HSR projects and suggests ways to reduce such risk. Combining a traditional risk management framework with a well-designed BN model adapted from Van Truong Luu [48], we propose a research framework of four phases. First, we identify risks by studying typical cases of HSR social risk. Second, based on the risk list, the causal link between risks is determined through case studies as well as expert interviews to build the BN structure. To determine the parameters of the risks, both expert interview and expert grading method are used. The BN model is thus developed and then applied to two case studies for validation purposes. Third, using the reverse inference and sensitivity analysis functions of Genie2 (a BN software program), the key risk factors, maximum causal chain, and sensitive risk factors are obtained. Finally, risk-control countermeasures are outlined to address these key points (Figure 2).

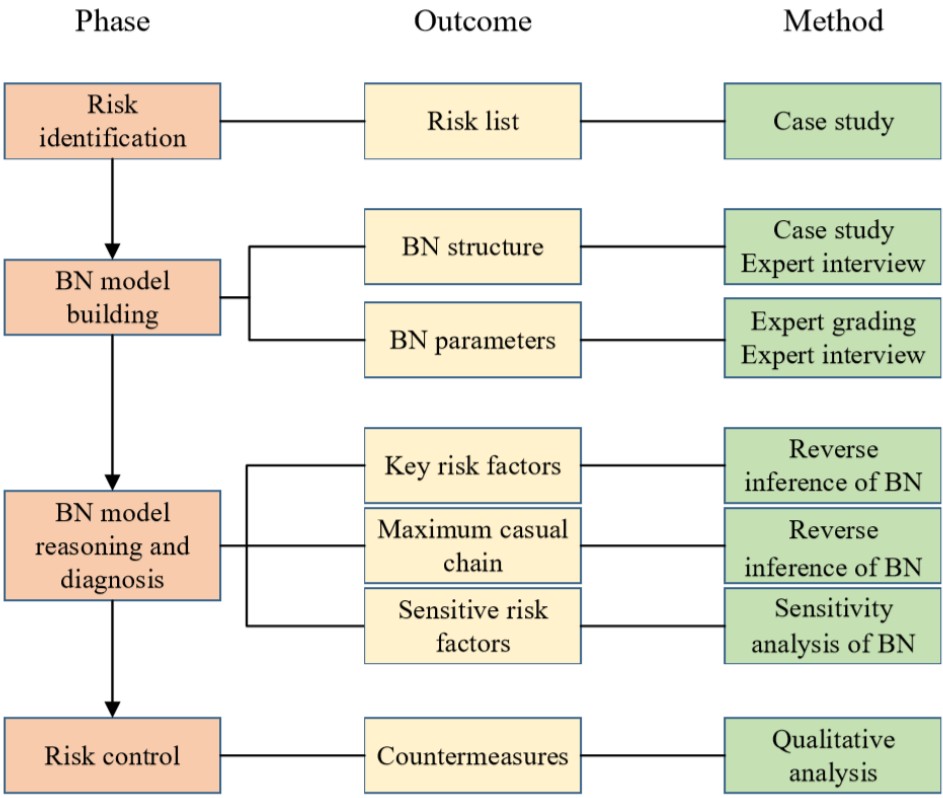

**Figure 2.** Research framework.

## 4. Research Process

### 4.1. Risk Identification

To accurately identify the social risks of HSR projects, typical cases first need to be selected and analyzed. Through the Internet, newspapers, and related books, we have collected news reports, expert comments, and video materials about a number of cases of social risk related to HSR projects. To select typical cases from these data, four criteria were eventually established after repeated discussions. The criteria are as follows:

1. The case needs to involve events that happened no more than a decade ago.
2. It needs to have been reported by mainstream media, such that data about the case are readily available.
3. It needs to have caused widespread social concern and had a long-lasting effect on public opinion.
4. It needs to have been associated with an individual incident or mass incident—that is, an incident in which a large group of people participated.

Based on this set of criteria, 10 typical cases of HSR projects were selected for risk identification; they are listed in Table 1. To make it more intuitive, the routes of these HSR projects and administrative regions participating in the HSR competition are shown in Figure 3 of China's 2020 HSR network.

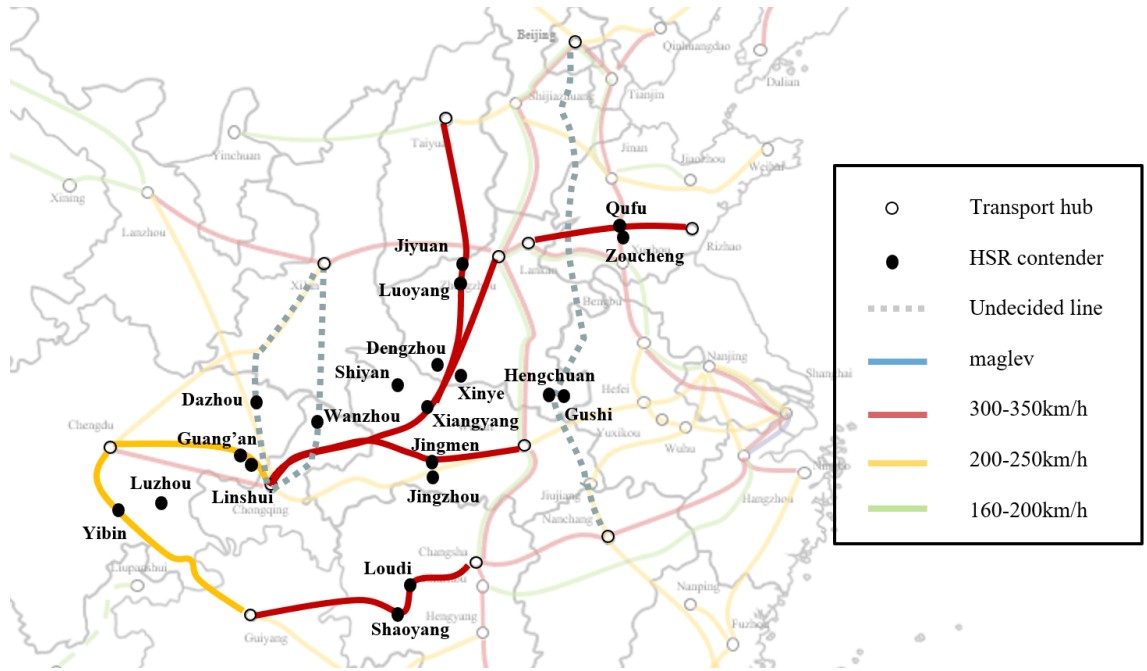

**Figure 3.** Sketch map of typical HSR lines and contender regions (2020).

By combining the relevant concepts from Section 2.2 with the research background for our study of social risk for HSR projects, we formulated the following definitions. The internal and external factors (subjective factors and objective factors) in the cases listed in Table 1 are regarded as risk factors for the social risk associated with the HSR projects. The key decisions or actions of the stakeholders leading to the actual losses (vs. potential risks) are called risk events and the actual losses, including the diminishment of government credibility, damage to personal safety damage, and property loss, are called risk consequences.

The specific operation of identifying risks via the case studies is divided into three steps:

Step 1. Sort out the phenomena, events, actions, and results in the data. For convenience, these elements are collectively referred to as risk performance.

Step 2. Summarize the risk factors, risk events, and risk consequences associated with risk performances. Thus, taking the Linshui 5/16 incident caused by the Dazhou-Chongqing HSR project as an example, in Step 1, nine risk factors, three risk events, and one risk consequence can be extracted (Table 2). In this way, risk identification is performed for each case.

Step 3. Integrate the risks extracted from each case in Step 2. In all, 20 risk factors, three risk events, and one risk consequence were obtained through the process of risk identification. Depending on the source of the risk in question, risk factors and risk events are classified into three categories: those associated with the planning system, those related to the local government, and those pertaining to the local population. For example, risk factors such as "Insufficient sharing of information" and "Demands of different interest groups not being taken into account" come from the planning system. Risk factors such as "Rent-seeking efforts by the local government" and "Fierce intergovernmental competition" come from the local government. Finally, risk factors such as "The local population's sense of injustice and relative deprivation" come, clearly, from the local population.

**Table 1.** Typical cases of social risk for high-speed railway (HSR) projects.

| Project Name | HSR Contender (Indicated by the Name of the Administrative Region) | Behavior of Local People | Response of Local Government | Initial Decision of HSR Station | Final Decision of HSR Station |
|---|---|---|---|---|---|
| Shanghai–Kunming HSR | Shaoyang, Loudi | Demonstrations and petition signing; displaying banners and demonstrations against local government | Supporting the demonstrators and actively striving to capture the project | Station in Loudi | Stations in both cities |
| Zhengzhou–Chongqing HSR (1) | Shiyan, Xiangyang | Demonstrations and large-scale petition signing; online voting | Supporting the demonstrators and actively striving to capture the project | Undecided | Station in Xiangyang |
| Zhengzhou–Chongqing HSR (2) | Dengzhou, Xinye | Setting up nongovernmental organizations; demonstrations and petition signing; displaying banners all over the area | Supporting the demonstrators and actively striving to capture the project | Station in Dengzhou | Station located between the two cities |
| Shanghai–Hangzhou–Rong River HSR | Jingzhou, Jingmen | Organizing initiatives in multiple locations; petition signing (more than 10,000 signatures) | Following the planning directives of the superior Planning Department | Undecided | Station in Jingmen |
| Dazhou–Chongqing HSR | Linshui, Guang'an | Mass demonstrations; displaying banners and demonstrations against government officials; petition signing (more than 10,000 signatures); conflicts between police and civilians | Supporting the demonstrators and actively striving to capture the project | Undecided | Stations in both cities |
| Chongqing–Xi'an HSR | Wanzhou, Dazhou | Massive social network campaigns and disputes; massive use of online voting | Supporting the demonstrators and actively striving to capture the project | Station in Wanzhou | Undecided |
| Beijing–Kowloon HSR | Gushi, Huangchuan | Petition signing (more than 10,000 signatures); displaying banners all over the area | Supporting the demonstrators and actively striving to capture the project | Station in Huangchuan | Undecided |
| Chengdu–Guiyang HSR | Luzhou, Yibin | Petition signing (more than 10,000 signatures); "netizens" write to the ministry of railways; massive social network campaigns and disputes | Supporting the demonstrators and actively striving to capture the project | Undecided | Station in Yibin |
| Huhehaote-Nanning HSR | Jiyuan, Luoyang | Large-scale petitions organized online | Supporting the demonstrators and actively striving to capture the project | Undecided | Station in both cities |
| Rizhao–Lankao HSR | Zoucheng, Qufu | Smaller-scale petitions organized online by individuals | Following the planning directives of the superior Planning Department | Undecided | Station in Qufu |

**Table 2.** Risk identification for the Dazhou–Chongqing HSR project.

| Risk Factor/Event/Consequence | Risk Performance | Name |
|---|---|---|
| Risk factor | Before being classified as part of Guang'an, Linshui had a larger population and a more developed economy than Guang'an. After Linshui was classified as part of Guang'an, the pace of its economic development and the quality of its former residents' welfare were not as good as they were for the long-term residents of Guang'an, resulting in discord between the former residents of Linshui and the original Guang'an people. | Regional conflicts |
| | In the master plan of Linshui County 2009–2030, Linshui County is positioned as the transportation hub of eastern Sichuan and northern Chongqing. Railway lines and railway stations have been planned for the southeast portion of the county. However, there is no railway in Linshui County currently. | Incompatibility between regional and local levels of planning for economic development |
| | People believe that the railway can utilize the geographical advantages of Linshui County and improve what is perceived as the economic backwardness of Linshui. Therefore, the local population has built up large expectations for the Dazhou-Chongqing HSR. | The local population's desire for transportation facilities |
| | Officials of the Guang'an municipal government visited the superior Planning Department and demanded that the route of the Dazhou-Chongqing HSR be changed to Guang'an. | The influence of rent-seeking by the local government |
| | Individuals set up "road-protecting" QQ groups to divide up online the tasks undertaken in support of the HSR. | The influence of civic organizations and group leaders |
| | Sentence in the petition: "We don't want a few people to determine the historical fate of the two counties off the top of their heads. Instead, a fair, carefully reasoned decision is urgently needed." | The local population's sense of injustice and relative deprivation |
| | Leaders of Guang'an City state that they will only accept the western line plan. At the same time, leaders of Linshui County are also striving to capture the project. | Fierce intergovernmental competition |
| | In the process of HSR planning, there is no effective social risk assessment and no formal way for people to express their demands. | Poor public access to participation in the planning process |
| | A total of 68 people (including police and civilians) are injured, several vehicles are burned, and property is lost. The credibility of the local government is seriously diminished. | Insufficient emergency management capacity of local government |
| Risk event | Some people wrote letters and paid visits to the county party committee and county government. Some initiated petitions on community websites such as "Spicy Community." On May 12, a small-scale petition signing initiative was launched in Huangtongshu Park. | Local population involved in an individual incident |
| | Over the next two days, neither the petition on the Internet nor the petition signing event was responded to by the government. | Local government fails to provide an immediate response to the individual incident |
| | On May 14, the voices of those organizing demonstrations and expressing appeals on the streets became mainstream. On May 16, tens of thousands of people gathered and marched in the streets. Later, police–civilian clashes occur and traffic trunk roads were blocked. | Local population involved in a mass incident |
| Risk consequence | A total of 68 people (including police and civilians) were injured, several vehicles were burned, and property was lost. The credibility of the local government was seriously diminished. | Social instability |

The above steps enable the compilation of an accurate risk list detailing risk factors, risk events, risk consequences, and their corresponding sources. Table 3 details the risk list highlighting the social risks that stem from HSR projects in China.

**Table 3.** Risk list.

| Risk Factor /Event/Consequence | Sources of Risk | No. | Name |
|---|---|---|---|
| Risk factor A | Planning system | A1 | Insufficient sharing of information |
| | | A2 | Poor public participation |
| | | A3 | Demands of different interest groups not being taken into account |
| | | A4 | Obvious bias of decision-makers |
| | | A5 | Incompatibility between regional and local levels of planning for economic development |
| | | A6 | Negative public opinion |
| | | A7 | Blind competition among regions, industries, and groups |
| | Local government | A8 | Pursuit of political achievements/reputation |
| | | A9 | Fierce intergovernmental competition |
| | | A10 | Rent-seeking efforts by local government |
| | | A11 | Inadequate emergency management capability |
| | | A12 | Poor communication with the public |
| | | A13 | Free-ride tendency |
| | Local population | A14 | Regional conflict |
| | | A15 | Desire for transportation facilities |
| | | A16 | Distrust of local government |
| | | A17 | Sense of injustice and relative deprivation |
| | | A18 | Low level of education |
| | | A19 | Civic organizations and group leaders |
| | | A20 | Imitative and conformist behavior |
| Risk event B | Local population | B1 | Individual incident |
| | Local government | B2 | No immediate response to individual incident |
| | Local population | B3 | Mass incident |
| Risk consequence C | - | C | Social instability |

*4.2. BN Model Building*

4.2.1. Establishing the BN Structure

According to the generation mechanism discussed previously, social risk is transmitted according to a chain path of "risk factor → risk event → risk consequence." If social risk factors are not controlled in the early stage, the risk energy will start to accumulate. When that energy is amplified to a certain degree, it can lead to risk events. When analyzing our case studies, we found that during the generation process of social risk for HSR projects, there is often more than one risk event. In other words, the risk events of social risk are a series of multi-stage dynamic occurrences involving an interplay between local populations and local governments (see Figure 4).

During the first stage, given a prior probability of relevant risk factors, when the risk energy accumulates beyond the tolerance threshold of the local population, they, as the first movers, decide whether to take action or not. In the cases we examined, the action involves a sequence of individual incidents, such as an individual organizing a petition, writing letters, visiting local officials, and initiating mobilization online.

During the second stage, as the second mover, the local government, observing the action of the local population, decides whether to immediately formulate an emergency response or not, to prevent or at least defuse individual incidents—given the prior probability of risk factors. The theory of risk-society expansion holds that the response of risk managers to information flow can either

strengthen or reduce people's perception of risk and shape their behaviors accordingly, in a manner that may, in turn, bring about new social or economic consequences. These consequences may far exceed the direct harm of the risk event itself, when it comes to human health or the health of the environment, leading to further important indirect effects, and even spreading to society at large [51]. If the government engages in inappropriate behavior or fails to address the issues at hand, then it may, in conjunction with the impact of related risk factors, cause further expansion of the risk.

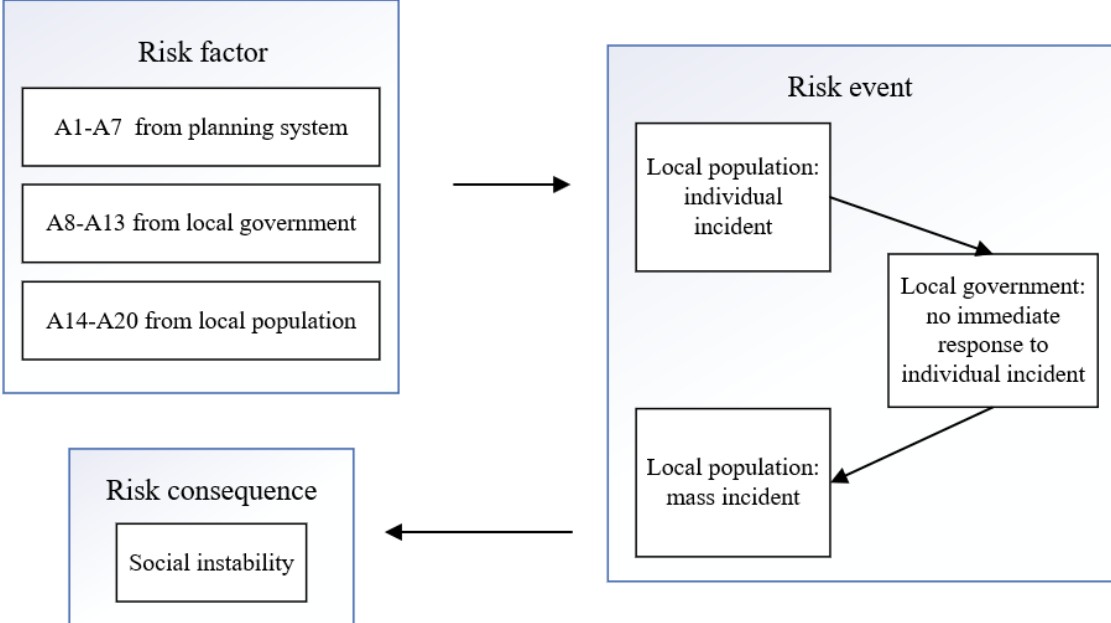

**Figure 4.** The path of social risk generation for HSR projects in China.

During the third stage, the prior probability of the risk factors is given and the local population has already observed the emergency response, or lack thereof, on the part of the local government. At this point, the local population decides whether to end the protest or take more drastic actions, such as mobilizing more groups to expand social influence. In the cases we examined, it was at this point that what were individual incidents in the first stage expanded to mass incidents, including mass petition signing drives, traffic congestion, street demonstrations, and conflicts between civilians and police. Once a risk event has evolved into a large-scale mass incident, it invariably gives rise to extensive concern and negative public opinion, greatly damaging the political authority of the government. It may also cause property loss and casualties along with more widespread violence. For these reasons, in China, it is generally believed that the occurrence of large-scale mass incidents will inevitably lead to the risk consequence of social instability [52].

The causal relationships among risk factors, risk events, and risk consequence are the path of risk generation for Chinese HSR projects. This path mirrors the structure of the BN that we use to model the mechanism for risk generation in this context. Based on the risk-generation model and the risk list we created based on the case studies that experts helped us analyze, we posited 25 sets of causal relationships such as "Insufficient sharing of information → Local population: individual incidents," thereby establishing the structure of the BN of social risk generation for Chinese HSR projects. Figure 5 provides a visualization of our model.

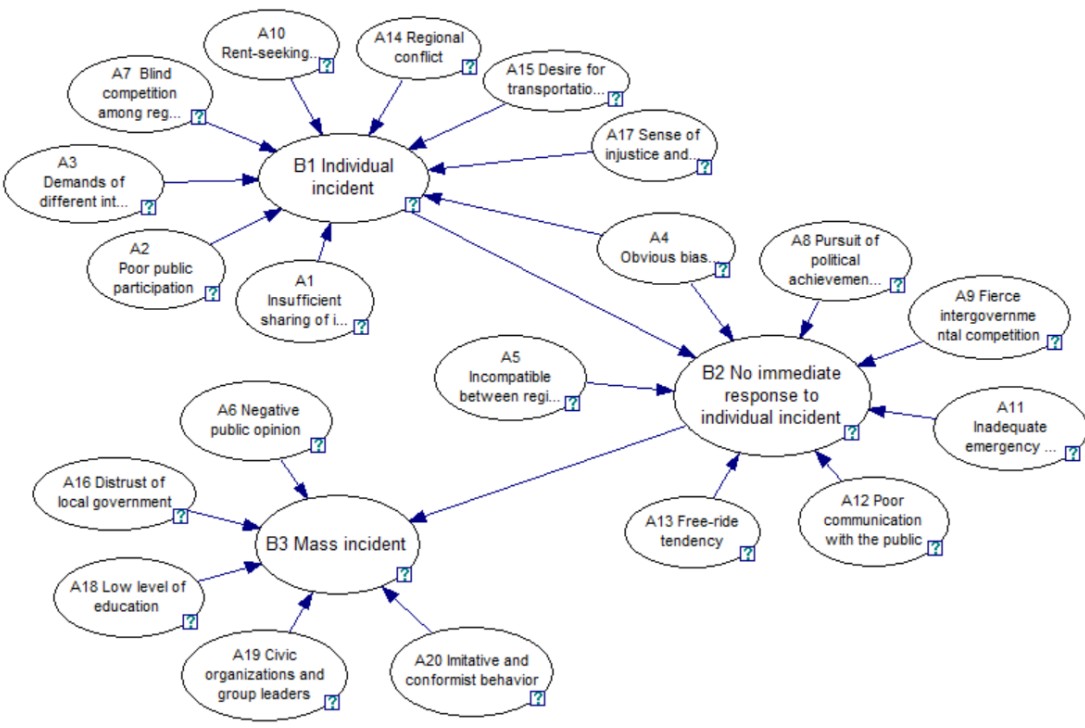

**Figure 5.** Bayesian network (BN) structure of social risk generation for HSR projects in China.

### 4.2.2. Determining the BN Parameters

To complete our BN model of the mechanism by which social risk is generated for HSR projects, we had to determine the BN parameters. In China, mass social incidents are highly politically sensitive, one-off (difficult to reproduce), and atypical. The data regarding such incidents in China is thus highly limited. Therefore, we used the expert grading method to determine the BN parameters.

As the basis for expert grading, we must find a way to assess the risk level of each of the risk nodes in the BN structure. According to the classic risk theory, the risk level can be measured by a function of likelihood and consequence [16,53]. Therefore, a questionnaire using the five-point Likert scale was designed for evaluating the likelihood of the occurrence and severity of its consequences for each risk node. Based on the practical experience of the experts of mega projects in China, the likelihood of occurrence scale includes scores ranging from 1 to 5, with 1 being "low" and 5 being "very high" (Table 4). Drawing on relevant national policy documents, the severity of consequence scale refers to the losses caused by social risk. It, too, includes scores ranging from 1 to 5, with 1 being "negligible" and 5 being "very serious" (Table 5). With each risk node having been evaluated along these two dimensions, a two-dimensional risk matrix was established. This matrix allowed the risk level of each risk node to be divided into low, medium, and high (Figure 6).

**Table 4.** Scale for the likelihood of occurrence.

| Questionnaire Value | 1 | 2 | 3 | 4 | 5 |
|---|---|---|---|---|---|
| **Name** | low | medium low | general | high | very high |
| **Probability** | 0–0.2 | 0.2–0.4 | 0.4–0.6 | 0.6–0.8 | 0.8–1 |

The parameters in the BN model include the prior probability of risk factors and the conditional probability table (CPT) of risk events. First, we determined the prior probability of risk factors by sending questionnaires to selected experts. There were three sections in the questionnaire: Section 1 introduced the background of the questionnaire and the specifics of each individual case to ensure that

respondents gained a thorough understanding of the social risks of each of the HSR projects described previously. Section 2 provided a detailed introduction of the assessment criteria in Tables 4 and 5, and in Section 3, respondents were presented with the social risk assessment that sums up the risk factors associated with the HSR projects, including the likelihood of each of the risk factors occurring and the severity of the risk consequences. A total of 22 experts in the field were invited to complete the questionnaire and 21 valid questionnaires were returned. Experts consisted of the following: professors who have undertaken extensive research on risk management and social conflicts from universities belonging to Project 985 in China; PhD members of the research team working on the present study; senior project managers from the China National Railway Corporation and China Railway Group, Ltd.; and senior managers in engineering consulting enterprises that are ranked in the top 10 in China. They were all proficient in social risk management and familiar with HSR project practice. Thus, we made every effort to ensure that the questionnaire data provided by the experts were credible and authoritative. After the questionnaire data were standardized, the data on risk levels were inputted into the BN using the Access software program to obtain the prior probability of each risk factor.

**Table 5.** Scale for the severity of consequences.

| Questionnaire Value | Name | Detailed Description |
|---|---|---|
| 1 | negligible | No (or little) public opposition<br>No negative public opinion or effect on governmental authority<br>No property loss and no injuries |
| 2 | less | Individual opposition that can be resolved through effective work<br>Minimal negative public opinion and effect on governmental authority<br>Little property loss and no injuries |
| 3 | medium | Small-scale conflicts that can be immediately alleviated via on-site management<br>Little negative public opinion and effect on governmental authority<br>Moderate property losses and individual injuries |
| 4 | serious | Medium-scale conflicts that require external assistance to mitigate<br>Considerable negative public opinion and effect on governmental authority<br>Large property losses and some casualties |
| 5 | very serious | Large-scale mass incidents or extreme incidents involving individuals<br>Widespread negative public opinion and significantly negative effect on governmental authority<br>Extensive property damage and numerous casualties |

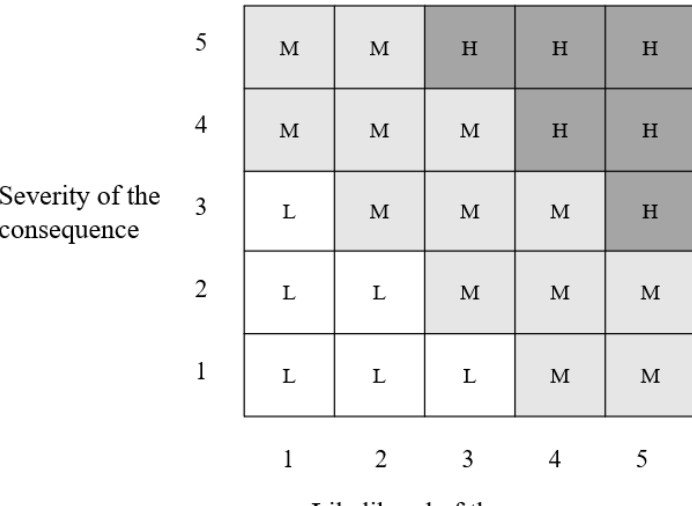

**Figure 6.** Two-dimensional matrix for risk assessment.

Then, to obtain the CPT of each risk event node (B1, B2, and B3), discussions were conducted with three top experts in the field of social risk management. Based on relevant materials, these experts evaluated the risk levels of each node in each case in Table 1. Grading data were again standardized and then inputted into GeNIe2 for parameter learning by the EM algorithm to generate CPTs of the risk event nodes. Due to the small number of typical cases in this study, the CPTs were modified manually. However, in the BN structure, the risk event nodes (B1, B2, and B3), as originally represented, had many parent nodes and therefore were very complex CPTs, raising the problems of inconvenience and time-consuming revisions. To address these problems, we changed the node type from "general" to "Noisy-MAX." Taking the B1 node with three risk states and nine parent nodes as an example, when its node type is "general," $3^9$ conditional probabilities are required to complete its CPT. By changing its node type to "Noisy-MAX," the multi-valued Noisy-OR gates proposed and improved by Diez [54] and Srinivas [55] can help model interactions among variables with multiple states and allow for specifying these interactions with few parameters [56]. By undertaking this step, the number of required conditional probabilities were reduced to the number of a given node's parents [49]. To ensure accuracy, repeated discussions were conducted and node parameters with obvious problems were corrected to the average values provided by the three experts. Once the probability distributions of all a child node's parent nodes have been established, the GeNie2 software application will automatically calculate the CPT of that child node. Through these procedures, we obtained the CPTs of all the risk event nodes (B1, B2, and B3). After the prior probability of the risk factors and the CPTs of risk events were inputted into the network, a complete BN model was generated.

### 4.2.3. Model Validation

To ensure the validity of the model, two cases with different risk consequences were used for model verification. The first case, the Dazho–Chongqing HSR project (Case A) was a large-scale mass incident, which resulted in 68 people being injured and incidents of criminal damage. In contrast, the Rizhao–Lankao HSR project (Case B) experienced relatively few protests, with only individual online petitions and no mass incident. By inputting the data of the two cases (from the grading process of the top experts mentioned above) into the model, we obtained the results shown in Figures 7 and 8. From Figure 7, in case A, we see that the probability of a high-risk occurrence of B3 (mass incident) is 79%. Therefore, mass incidents are more likely to occur in such situations, leading to social instability. From Figure 8, in case B, we see that the probability of a low-risk occurrence of B3 (mass incident) is 71%; therefore, that social instability is much less likely. The simulation results of the two cases are clearly different and are consistent with real-world occurrences; thus, the BN model is effective and can be used as the basis for subsequent inference and analysis.

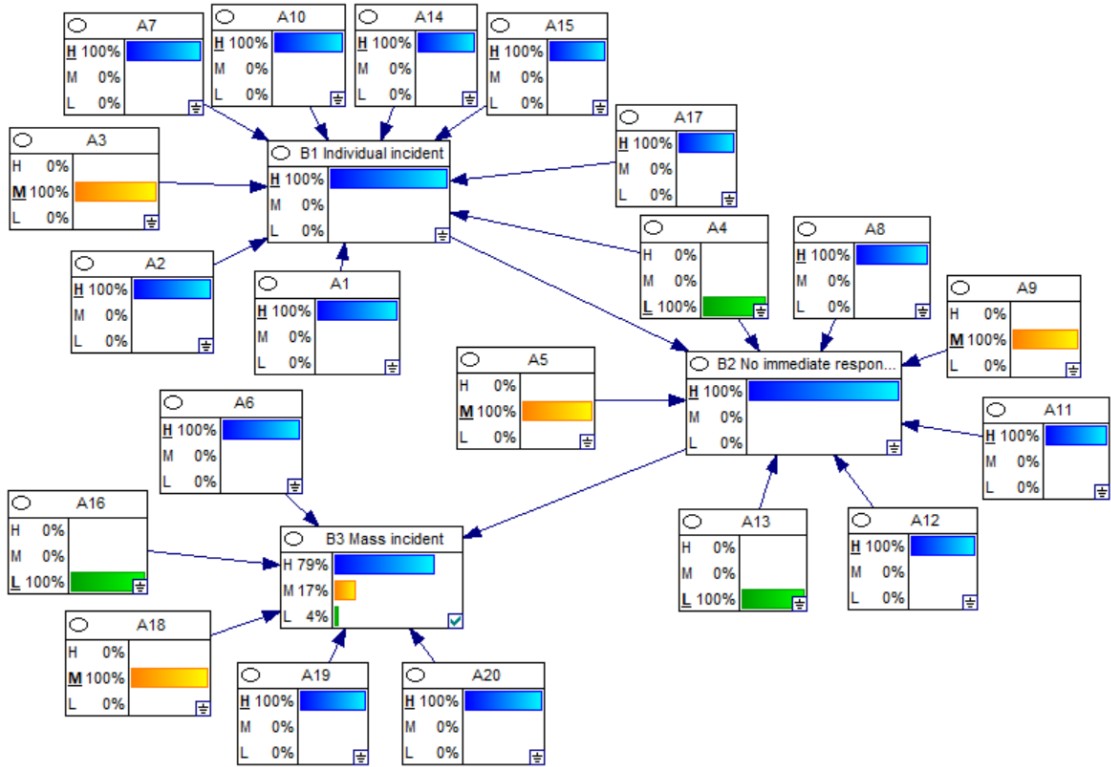

**Figure 7.** The BN simulation result of case A.

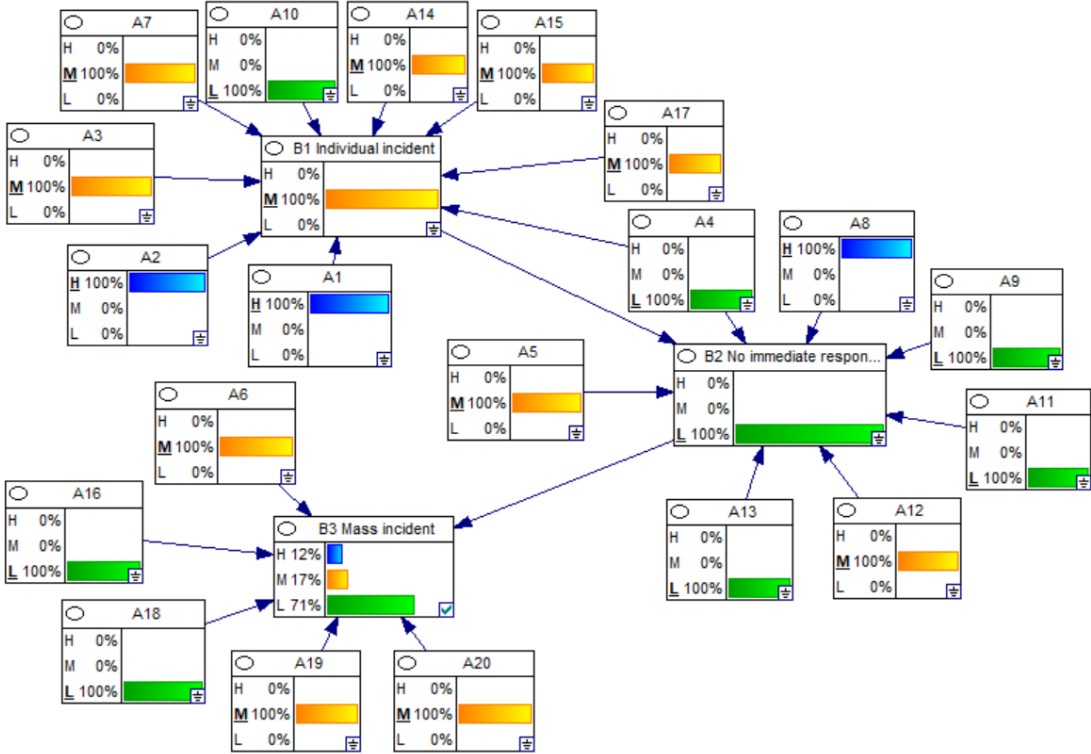

**Figure 8.** The BN simulation result of case B.

## 5. Results

BN models have powerful capacities for inference and diagnosis, and the BN applications used in the present study include reverse inference and sensitivity analysis. Reverse inference, also called target-driven or hypothesis-driven reasoning, is useful for diagnosing and explaining the cause of an accident in a manner that runs counter to a directed graph—that is, graph modeling an accident whose result is already determined [57]. Sensitivity analysis refers to a technique for conducting uncertainty analysis; the technique sifts out from several influential factors the sensitive factors that have an especially important impact on the subject. Due to the reverse inference and sensitivity analysis capacities of the BN model, the regularities underlying the generation of social risk can be identified.

### 5.1. Reverse Inference

### 5.1.1. Key Risk Factors

As mentioned in Section 4.2.1, in China, the occurrence of large-scale mass incidents is believed to lead to the risk consequence of social instability. Based on the reverse inference algorithm in BNs, the high-risk state of the "Local population: mass incidents" node is set to 100%, which means that social instability is inevitable. Additionally, the percentages of H, M, and L states represent the probability of risk factors/events at different levels of risk when consequences (instances of social instability) occur. Factors with a high-risk state greater than 50% are called key risk factors and are marked yellow Figure 9. Figure 9 shows that the key risk factors include A1, A2, A8, A9, A17, A19, and A20. When one or more of the key risk factors occur, the possibility of social instability is relatively high.

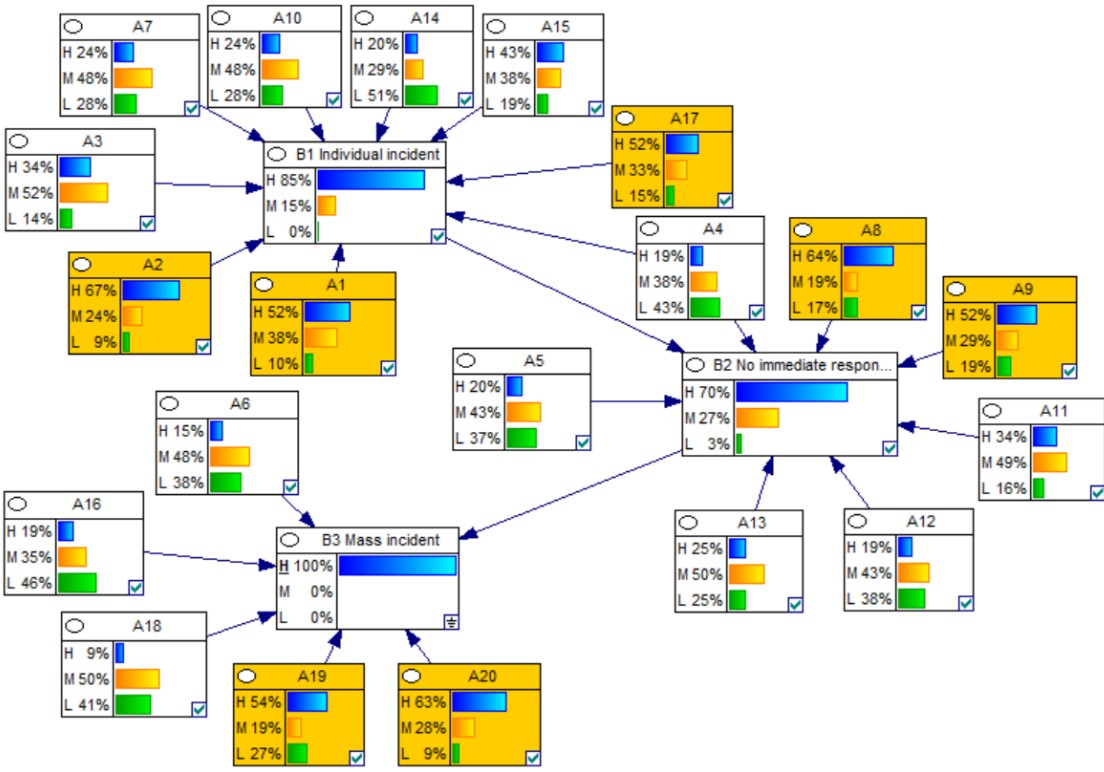

**Figure 9.** Reverse inference result and key risk factors.

### 5.1.2. Maximum Causal Chain

In the present study, a maximum causal chain refers to the most likely path that causes the risk consequence. By using the "strength of influence" tool under "network" in GeNie2, the maximum causal chain corresponding to our BN analysis can be generated; it is shown in bold in Figure 10.

The interaction between the local population and the local government, that is, the dynamic evolution of risk events of the form B1 → B2 → B3, is the maximum causal chain of social risk. Combined with Figure 9, when a high-level mass incident (B3) occurs, the probability of a high-risk individual incident (B1) is 85%, and the probability that the local government will not handle the individual incident (B2) in a timely fashion is 70%. The results of this analysis are basically consistent with the facts in the cases under study.

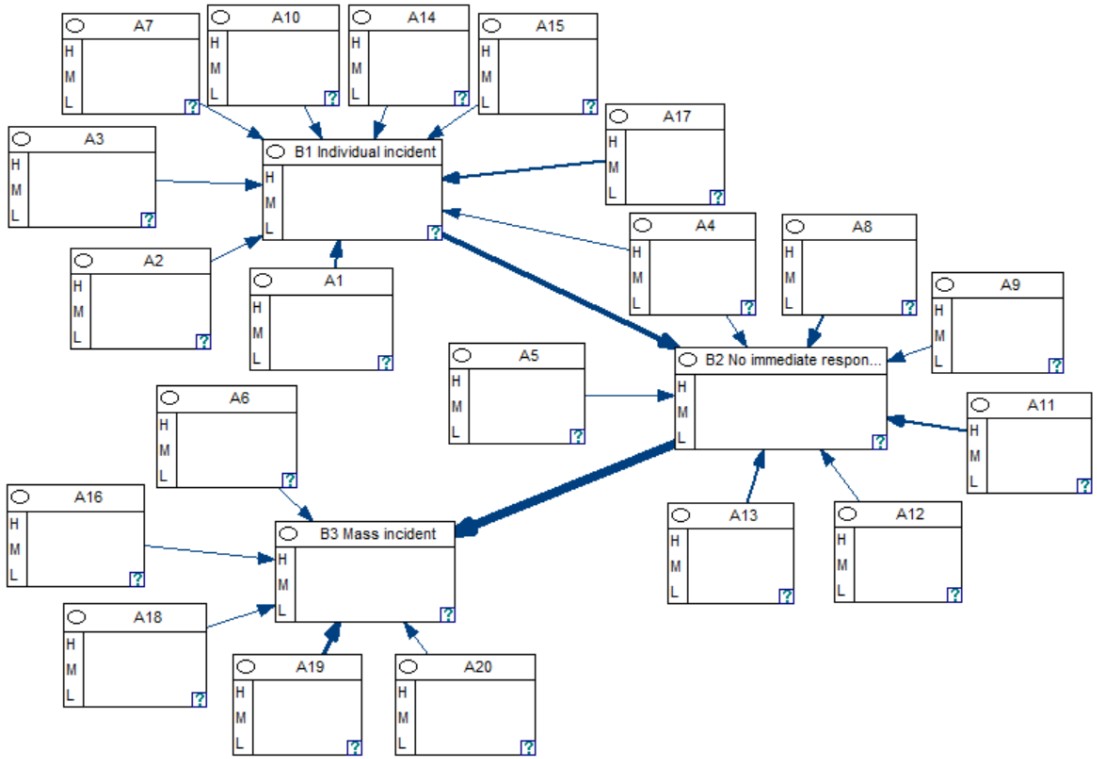

**Figure 10.** Maximum causal chain result in BN.

## 5.2. Sensitivity Analysis

In addition to the key risk factors and the maximum causal chain, the sensitive risk factors also need to be considered. Although the absolute probability of the occurrence of such factors is smaller than that of the key risk factors, the factors have a considerable impact on other nodes. By identifying and monitoring sensitive risk factors, we can help managers prevent hidden risks and reduce losses.

To identify the sensitive risk factors, GeNIe2 implements an algorithm proposed by Kjaerulff and Van der Gaag [58], that performs a simple sensitivity analysis in BN. Given a target node, the algorithm efficiently calculates derivatives of the posterior probability distributions over the target node for each of the numerical parameters of the BN [59]. These derivatives highlight the importance of developing precise network numerical parameters for calculating the posterior probabilities of the target, so we call them sensitive values [56,60]. Based on the results of reverse inference, with the "Mass incident" node (B3) set as the target, the sensitivity analysis tool in Genie 2 is used to analyze the sensitivity of each risk factor node (i.e., nodes A1–A20). At the end of this process, we rank the risk factors according to their sensitivity values (Table 6), and, based on the principle of 80/20, take the top 20% (nodes A14, A13, A6, and A4) as the sensitive risk factors. The results are shown in Table 6.

Hence, using the reverse inference and sensitivity analysis capacities of the BN model and combining these capacities with the sources of risk detailed in Table 3, we summarize the results given by the model regarding the social risks of the HSR projects (Table 7).

**Table 6.** Sensitive values of risk factors for "mass incident".

| Rank | Risk Factor | Sensitive Value | Rank | Risk Factor | Sensitive Value |
|------|-------------|-----------------|------|-------------|-----------------|
| 1 | A14 | 0.742 | 11 | A16 | 0.101 |
| 2 | A13 | 0.403 | 12 | A2 | 0.074 |
| 3 | A6 | 0.352 | 13 | A12 | 0.042 |
| 4 | A4 | 0.267 | 14 | A1 | 0.041 |
| 5 | A3 | 0.244 | 15 | A20 | 0.039 |
| 6 | A5 | 0.207 | 16 | A8 | 0.029 |
| 7 | A7 | 0.205 | 17 | A17 | 0.029 |
| 8 | A10 | 0.205 | 18 | A15 | 0.025 |
| 9 | A11 | 0.191 | 19 | A9 | 0.024 |
| 10 | A18 | 0.154 | 20 | A19 | 0.020 |

**Table 7.** Summary of results.

| Sources of Risk | Key Risk Factor | Sensitive Risk Factor | Link in the Maximum Causal Chain |
|-----------------|-----------------|-----------------------|----------------------------------|
| Planning system | A1 (Insufficient sharing of information) <br> A2 (Poor public participation) | A6 (Negative public opinion) <br> A4 (Obvious bias of decision-makers) | - |
| Local population | A17 (Sense of injustice and relative deprivation) <br> A19 (Civic organizations and group leaders) <br> A20 (Imitative and conformist behavior) | A14 (Regional conflict) | B1 (Individual incidents) <br><br> B3 (Mass incidents) |
| Local government | A8 (Pursuit of political achievements/reputation) <br> A9 (Fierce intergovernmental competition) <br> A20 (Imitative and conformist behavior) | A13 (Free-ride tendency) | B2 (No immediate response to individual incident) |

Therefore, in project management, it is necessary to focus on the risk factors and risk events, as shown in Table 7. It is also imperative to take risk-control countermeasures that limit the likelihood of occurrence of risks, and thereby reduce the potential for risk losses.

## 6. Discussion of Results and Description of Countermeasures

As part of our discussion of these research results, we turn now to a brief discussion of risk-controlling countermeasures. The government is the main risk manager of social risk in China [15]; therefore, these three countermeasures are mainly formulated for potential adoption by the government.

*6.1. First Countermeasure: Increase the Transparency of and Democratize HSR Planning*

Based on the results presented in Table 7, several problems affect the process of HSR project planning, including insufficient sharing of information and poor public participation. Moreover, when decision-making officials have biased interests or the planning scheme leads to negative public opinion, these shortcomings are likely to lead to mass incidents. There are two explanations for this pattern. First, in China, HSR project planning is a relatively closed decision-making process limited to experts and these decision-makers lack a mechanism for disclosing information to the public. Hence, changes in planning schemes often generate public opinion utilizing dialogue conducted through unofficial media and, consequently, public opinion rarely receives a positive official response. As the stakeholders of HSR projects, the public is understandably anxious about this information asymmetry. Second, social-impact evaluations of planned HSR projects are currently conducted via Social Stability Risk Assessment. However, this method of risk assessment fails to consider public risk perceptions [61] as key points of "concern assessment" [62]. There is thus little chance the method can prevent the effects of social risk.

In this regard, relevant government departments should start by improving the way information regarding HSR schemes is disseminated, to ensure that this major administrative decision is well

publicized—and well-understood by the public. Through government websites, government media, newspapers, radio, television, and other means that are accessible to the public, the basis, process, latest progress, and results of planning schemes can be disclosed in a timely, transparent manner. At the same time, the summary of public opinions and official risk assessments of the project should be disclosed. Second, and with the help of this improved information disclosure, high-quality public participation and engagement should be encouraged. Drawing on the experience of the British Railway Network Company's Long-Term Planning Process standard [63], while adjusting it for the current cultural and technological conditions in China, we propose three suggestions concerning public participation in planning for HSR projects. First, the local government should provide the public with official online channels to generate more ideas and views. Second, more emphasis should be placed on the nongovernmental organizations that act as mediators between the public and government. Third, the social impact of the planning process should itself be well tracked and evaluated for a period. In these ways, the public can express their opinion in a more civilized and equitable manner, thus defusing social risk.

*6.2. Second Countermeasure: Improve the Mechanism by Which Local Governments can Express Interest in HSR Projects*

As our analysis suggests, the local population is only one source of the social risk of HSR projects. Further, as indicated in Table 7, the motives and behaviors of local governments are also responsible. In China, the construction of the HSR project is jointly financed by the central government and local governments; local governments inject capital in the form of land, and thus they have their say in the planning of the railway. For example, in the case of the Zhengzhou–Chongqing HSR project, the Henan provincial government has repeatedly put forward suggestions for revisions to the plan and conducted discussions with the railway corporation and various experts about possible adjustments to the plan. In general, local government officials see HSR as a new economic growth point for their region, as they seek to increase the region's sustainable fiscal revenue. Securing HSR can also be instrumental for their own career advancement. Therefore, in the case of individual incidents, motivated by a free-ride tendency, local governments may choose to acquiesce to public protests to create a public-opinion atmosphere that attracts the attention of superiors, and thus helps them compete for the HSR station. It can be concluded that local governments can intervene in HSR planning through two main channels: (1) by making interest appeals to higher decision-making departments through formal or informal means; and (2) by expanding the influence of these interest appeals through an emphasis on the social risks brought by public discontent.

It is therefore advisable to find a formal approach or establish a legal system that allows local governments to make their interest appeals reasonably and legally and to ensure the effectiveness of those appeals without governments having to resort to the encouragement or at least passive acceptance of mass incidents. First, an organizational structure can be set up for purposes of expressing interest in HSR projects. Relevant decision-making departments of government and experts on HSR projects are the expression subjects and the project risk-control agencies, and acting independently from those expression subjects are the coordination subjects. Second, institutionalized interest expression procedures should be established. For example, local governments can integrate public opinion, economic performance data, and environmental reports, and then submit formal written documents to coordination subjects. To ensure the effectiveness of the system for these expressions of interest by local governments, government departments and experts need to make all these appeals and their associated responses available to the public. In the case of special circumstances, such as major differences between local governments, the coordination subject can organize coordination meetings.

*6.3. Third Countermeasure: Enhance Emergency Management Mechanisms*

As the key risk factors directly related to civic actions, "Sense of injustice and relative deprivation," "Civic organizations and group leaders," and "Imitative and conformist behavior" show the basic

logic of collective action. For Stephen P. Robbins, a group is a collection of two or more people with the same interests or emotions who are somehow related to each other [64]. Through their initial contact with members of the group via mass media, group leaders form their own set of cognitive systems and become public-opinion leaders to initiate group mobilization. In the emotions arising from this interaction, the group will form a kind of group consciousness. The members of the group will quickly and without judgment accept certain information and act accordingly, or subconsciously, by imitation, as they respond similarly to a stimulus [65]. Among potential stimuli, "Regional conflict" tends to cause strong individual dissatisfaction and helps create an emotional consensus; this process can quickly improve the cohesion of the group.

To cope with this dynamic, it is necessary to improve the emergency management mechanism used at various stages of risk events. At the stage of individual incidents, the local government should attach great importance to the opinions of the people, pay attention to consultation and social mediation, and promote a responsive system for handling letters and visits as well as a system for receiving visits by group leaders. In addition, to prevent further deterioration of the event, it is necessary to implement real-time monitoring of public-opinion dynamics to stay abreast of the potential for social conflicts and disputes. At the stage of mass incidents, it is necessary to ensure that the emergency management department has a reasonable organizational structure and personnel composition, and to prepare an emergency plan for mass incidents that can be quickly implemented. The command system should mobilize police according to the pre-set plan to control the situation and minimize social and economic losses caused by mass incidents [66]. After the event, accurate information should be released to the media, lessons from the event should be learned, and the whole experience should be concluded with all parties on a good or at least mutually acceptable footing. Emergency plans should also be revised as necessary.

## 7. Conclusions and Future

According to sociologist Zhao Dingxin, "in recent years, social struggle in China has evolved from a reactive type to an active type [67]." Because of their potential social and economic benefits, HSR projects lead to efforts by the public and local governments to capture such projects for their regions. In the past decade, there have been a number of related mass incidents in China, resulting in adverse social impacts. In this context, the present study explored the mechanism of social risk of HSR projects using a BN model, and, based on the analysis supported by that model, put forward risk-control countermeasures. Our research conclusions can help the government reduce social uncertainty in the planning of HSR projects, in a manner that is conducive to the long-term stability of the country. More broadly, with ongoing economic development and urbanization in China, many projects involving large-scale infrastructures with massive externalities are being planned. At the same time, the Chinese public's desire to participate in public decision-making processes related to their own interests has increased, due to their increased awareness of rights, including the right to self-expression. Therefore, similar problems are likely to arise around other infrastructure projects. Our findings can thus guide a wide range of social risk scenarios, going forward. The research process and methodology can be applied to the analysis of social risk in other countries, providing a profitable reference.

Our main conclusions are as follows:

- Through a risk-identification process drawing on typical cases, a risk list containing 20 risk factors, three risk events, and one risk consequence was created; in this list, the risks are divided into three levels according to their source. Based on the causal relationship among the risks identified by the study of typical cases and through interviews with experts, a BN structure was constructed. Using an expert-based grading system, parameters were assigned to each node. The BN model was then applied to two case studies for validation purposes. The results yielded a valid BN model of the generation of social risk for HSR projects.

- Results from the reverse inference and sensitivity analysis functions of the BN model indicated that nodes A1, A2, A8, A9, A17, A19, and A20 in the model were the key risk factors; that nodes A4, A6,

A13, and A14 were the sensitive risk factors; and that B1→B2→B3 was the maximum causal chain. The study then considered ways in which social risk might be effectively controlled. Bringing the above results to bear on the risk sources, including the planning system, local populations, and local governments, the paper proposed three countermeasures: namely, increase the transparency of and democratize HSR planning, improve the mechanism by which local governments can express interest in HSR projects, and enhance emergency management mechanisms.

There are two main limitations to this study. First, the research mainly depends on expert knowledge and is therefore somewhat subjective. This is a problem that also arises in research on China's emergency management and public-safety management systems [43]. Second, the research is conducted from a macro perspective, without considering the heterogeneity of the public's perception of risk at the individual level. Given these considerations, further studies should focus on two issues. First, a planning method that balances science and public opinion needs to be developed, to implement optimal—i.e., socially sustainable—decision-making procedures. Second, as big data platforms are gradually used in the field of emergency management, big data analysis of the behavioral patterns of large-scale and heterogeneous actors can be used to improve the objectivity of research in this area and help optimize social risk management.

**Author Contributions:** Conceptualization, Y.X. and P.X.; Funding acquisition, P.X.; Investigation, Y.X.; Methodology, Y.X.; Software, Y.X.; Supervision, P.X. All authors have read and agreed to the published version of the manuscript.

**Funding:** The work described in this paper is fully supported by a joint grant from Chongqing Social Science Foundation Project (2018ZD02) and Project 2019CDJSK03PT07, No. 2017CDJSK03XK19, supported by the Fundamental Research Funds for the Central Universities.

**Conflicts of Interest:** The authors declare no conflict of interest.

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
