# Peer review of "The Social Risk of High-Speed Rail Projects in China: A Bayesian Network Analysis"

_sustainability, doi:10.3390/su12052087_

Round 1

Reviewer 1 Report

There is no new approach, but this paper is well-written based on a logical flow. There are limitations due to lack of data, but I think it can be helpful to identify the social risks of High-speed Rail Projects. I think that it can be published if the authors supplement some parts.

1. line 59: Please add a reference to the event.
2. In this study, HSR project is analyzed. However, since no map is provided, it is difficult for those who do not know China's HSR system to grasp intuitively. So add the current HSR map of China as a figure. Also mark the projects listed in Table 1 with the region names and paths on the map so readers can see them in the figure.
3. The description of Table 3 is not specified in the text.
4. All analyzes in this study rely on expert opinion and have a poor statistical base. In my opinion, I want authors to list the number of cases where each risk actually occurred in the risk list of Table 3. If this process is not worth due to lack of data, I recommend that you verify the similarity between the analysis result and the actual case for a specific case. If the result does not show a significant correlation, it is difficult to trust this study. I think that it is very important in this paper.
5. What does the question mark mean in Figure 4? If the results from the software are displayed, please delete them. Or specify in the caption of Figure 4 that the figure is the result of a software analysis.
6. line 323: The content of the questionnaire is not explained at all in the text. Readers need to know if the survey is based on appropriate contents. Please fill out the survey contents for readers to understand.
7. line 334: The result of changing the node type from "general" to "Noisy Max" is explained. But readers can’t know what each one means. Supplement the explanation for each.
8. Line 345: Please add a reference for “Reverse inference”.
9. line 359: Please modify Figure 4 to Figure 6.
10. Figure 6: Please explain what each percent of H, M, and L means.
11. Table 7: Specify in the caption that this table is sensitive values for mass incidents. Table 6 is missing, so fix it to Table 6. Then modify Table 7 to Table 6 on line 392.
12. line 398-409: It is difficult to understand the analysis result intuitively. If possible, please summarize the results in a table.

Author Response

Response to Reviewer Comments 

Dear Editors and Reviewer:

Thank you very much for the editorial staff and reviewer’s hard-work in processing our manuscript (sustainability-724725). We have carefully studied the comments. Also, we have tried our best to revise our manuscript according to the reviewer’s suggestions, and tracked changes in the text for the reviewers’ convenience. We hope this revision can make our paper more acceptable. The comments are addressed below in a point-by-point manner.

Point 1: line 59: Please add a reference to the event.

Response 1: Thank you very much for your careful work. According to your suggestion, we have added a reference in line 61.

Point 2: In this study, HSR project is analyzed. However, since no map is provided, it is difficult for those who do not know China's HSR system to grasp intuitively. So add the current HSR map of China as a figure. Also mark the projects listed in Table 1 with the region names and paths on the map so readers can see them in the figure.

Response 2: Thank you for the nice advice. We quite agree with you that, in previous version of the manuscript, we did ignore that many readers may not know China's HSR system.

As your suggestion, we add Figure 3 (Line 223). Figure 3 contains China's HSR network in 2020 and the region names and routes of HSR projects listed in Table 1(described in Line 219-221). In this way, we try our best to let readers understand the background of our topic more intuitively.

Point 3: The description of Table 3 is not specified in the text.

Response 3: Thank you for your careful reading. We specified the description of table 3 in the text and improved the description, please see Line 250-252.

Point 4: All analyzes in this study rely on expert opinion and have a poor statistical base. In my opinion, I want authors to list the number of cases where each risk actually occurred in the risk list of Table 3. If this process is not worth due to lack of data, I recommend that you verify the similarity between the analysis result and the actual case for a specific case. If the result does not show a significant correlation, it is difficult to trust this study. I think that it is very important in this paper.

Response 4: Thank you for the comments about the issue, we agree with you that it is important in this paper. As mentioned in the paper, due to the peculiarity of social events in China, accurate data are difficult to obtain. Follow your next suggestion, we added section 4.2.3 for model validation in model building process. Two actual cases with completely different risk consequences were used for model verification, and the simulation results of two cases are clearly different and consistent with real-world occurrences. Therefore, the BN model is considered to be effective and can be used as the basis for subsequent inference and analysis. Please see 4.2.3 Model validation (Line362...) for more details.

In addition, the key points and rules / mechanism in chapter “5 results” can be explained in accordance with reality in chapter “6 discussion and countermeasures”, which means that the model is credible. Thanks again for this insightful suggestion!

Point 5: What does the question mark mean in Figure 4? If the results from the software are displayed, please delete them. Or specify in the caption of Figure 4 that the figure is the result of a software analysis.

Response 5: Thank you for your careful work. The question mark is an initial state of the status icon. It means on this node, no evidence is set and no output value is displayed. Therefore, question marks are appropriate as status icons in figure 5(and, there is no option to delete them in the software).

Besides, it should be noted that we have not introduce status icons in detail in paper, because these details are explained in reference 57 (the user manual of software).

Point 6: line 323: The content of the questionnaire is not explained at all in the text. Readers need to know if the survey is based on appropriate contents. Please fill out the survey contents for readers to understand.

Response 6: Thank you very much for your valuable and thoughtful comments. According to your suggestion, we supplemented the detailed contents of the questionnaire in Line 324-330. In brief, the questionnaire consisted of three sections: Section 1 introduced the background of the questionnaire and the situation of the cases. Section 2 introduced the assessment criteria, and in Section 3, respondents were presented the social risk assessment, including the likelihood of occurrence and the consequence of each individual risk factor.

Point 7: line 334: The result of changing the node type from "general" to "Noisy Max" is explained. But readers can’t know what each one means. Supplement the explanation for each.

Response 7: Thank you for your valuable advice. "general" and "Noisy-Max" are two basic types of nodes. There is no distinction between the two types in the graph view, as they differ only in the way their conditional probability distributions (CPT) are specified. Taking an example of the B1 node, we supplemented the explanation of "general" and "Noisy-Max" in Line 349-353, please see them in revised manuscript.

Due to length limitations and the topic of this article, we did not expand on the specific principles of logic gate of "Noisy-Max", but added a reference of the user manual of software and related algorithms in sentences for interested readers.

Point 8: Line 345: Please add a reference for “Reverse inference”.

Response 8: Thank you for your careful work. As your suggestion, we add a reference for “Reverse inference” in Line 386.

Point 9: line 359: Please modify Figure 4 to Figure 6.

Response 9: Sorry for the error. We have corrected it (Line 399) and carefully rechecked the captions of other figures and tables.

Point 10. Figure 6: Please explain what each percent of H, M, and L means.

Response 10: Thank you for your valuable advice. We added an explanation in Line 396-398. The percentages of H, M and L states represent the probability of risk factors/ events at different levels of risk when risk consequences (instances of social instability) occur.

Point 11. Table 7: Specify in the caption that this table is sensitive values for mass incidents. Table 6 is missing, so fix it to Table 6. Then modify Table 7 to Table 6 on line 392.

Response 11: Thank you for your careful work. Caption of table 6 is replaced to “Sensitive values of risk factors for ‘Mass incident’”(Line 434). And the sequence numbers of Figures and Tables are modified and rechecked.

Point 12. line 398-409: It is difficult to understand the analysis result intuitively. If possible, please summarize the results in a table.

Response 12: Thank you! we totally agree with you and as your suggestion, the contents of result have been summarized in Table 7 (Line 439…). In this way, the results are more intuitive for readers.

We tried our best to improve the manuscript and made some other changes in the manuscript.

We appreciate for Editors/Reviewer’s warm work earnestly, and hope that the correction will meet with approval.

Once again, thank you very much for your comments and suggestions.

With best regards,

Sincerely yours,

Pengcheng Xiang

Dr. & Prof.

School of Management Science and Real Estate

Chongqing University

Chongqing P.R.China, 400045

Tel: 86-23-65120848

E-mail: [email protected], [email protected].

Reviewer 2 Report

The purpose of this paper is to study the mechanisms responsible for generating the social risk associated with high-speed rail projects. Based on a risk list and a group of typical case studies, a Bayesian network model is developed through case studies, expert interviews and expert grading. The topic is quite interesting and the methodology suitable for the research. Taken into account this positive assessment as a whole, I would like to mention some minor aspects (or reflections) of the paper in order to be published:

1)Introduction.

- Authors should remark in the introduction the high cost of new HSR lines (construction and operation). In Europe and the US, support for the new projects is facing serious concerns over the extremely elevated costs of high-speed and the ability of today's governments to fund or co-fund these systems. This is the main reason together to “social risk” for the assessment of methodologies to prioritise the construction of new high-speed rail (HSR) lines. Please, mention the works by Guirao and Campa (2014), proposed a ranking methodology to prioritize the construction of new lines

Guirao, B, Campa J.L. (2014). The construction of a HSR network using a ranking methodology to prioritise corridors. Land Use Policy, volume 38, May 2014, Pages 290-299.

- Authors write:…. “Especially for small and medium-sized cities, and also for poorer areas, their competitiveness will be strongly influenced by the quality of their transport connectionsto national and international supply chains and innovation networks, with benefits for a wide range  of local stakeholders.” It is not clear that small cities benefit from HSR arrival: we can find polarization and centralization processes. HSR operation (frequency and timetables) planned for new lines are also important. Ureña et al (2009) and Guirao et al (2018) have showed this approach. Please, mention their works:

Ureña J. M., Menerault P. and Garmendia M. (2009), 'The high-speed rail challenge for big intermediate cities: A national, regional and local perspective', Cities, 26, 5, 266-279.

Guirao, B, Campa J.L. and Casado-Sanz (2018). Labour mobility between cities and metropolitan integration: The role of high speed rail commuting in Spain. Cities, vol. 78, pp 140-154.

Guirao, B. (2013). Spain: highs and lows of 20 years of HSR operation. Journal of Transport Geography Volume 31, July 2013, Pages 201-206

3) Conclusions/results

Please mention the problems of transferability of this methodology (Bayesian network) and approach (social risks of proyects) o other countries different from China

Author Response

Response to Reviewer Comments

Dear Editors and Reviewer:

Thank you very much for the editorial staff and reviewer’s hard-work in processing our manuscript (sustainability-724725). We have carefully studied the comments. Also, we have tried our best to revise our manuscript according to the reviewer’s suggestions, and tracked changes in the text for the reviewers’ convenience. We hope this revision can make our paper more acceptable. The comments are addressed below in a point-by-point manner.

Point 1: Introduction

Authors should remark in the introduction the high cost of new HSR lines (construction and operation). In Europe and the US, support for the new projects is facing serious concerns over the extremely elevated costs of high-speed and the ability of today's governments to fund or co-fund these systems. This is the main reason together to “social risk” for the assessment of methodologies to prioritize the construction of new high-speed rail (HSR) lines. Please, mention the works by Guirao and Campa (2014), proposed a ranking methodology to prioritize the construction of new lines.

Response 1:

First, we do appreciate your favorable encourage and positive assessment as a whole. It has greatly encouraged my work and future research.

Then, we would like to thank you for the comments about the issue. According to your suggestion, we carefully read the relevant literatures and have a deeper understanding of the research background. As you mentioned, a new HSR lines generally face high cost in the construction and operation stage. If there is no predictable/ scientific decision during the planning phase, secondary risks such as local debt might be triggered in later stages. This is the main reason together to “social risk” for the assessment of methodologies to prioritize the construction of new high-speed rail (HSR) lines. The content is supplemented in introduction (Line 61-67).

Last but not least, thanks to the reviewers for the literatures, they are very valuable for the article and give us inspiration for further research. These literatures have appeared as references in our article.

Point 2: Authors write:…. “Especially for small and medium-sized cities, and also for poorer areas, their competitiveness will be strongly influenced by the quality of their transport connections to national and international supply chains and innovation networks, with benefits for a wide range of local stakeholders.” It is not clear that small cities benefit from HSR arrival: we can find polarization and centralization processes. HSR operation (frequency and timetables) planned for new lines are also important. Ureña et al (2009) and Guirao et al (2018) have showed this approach.

Response 2: Thank you very much for your valuable and thoughtful comments. We are very sorry for the imprecise description that small cities will benefit from HSR arrival. After reading the literatures carefully, we fully agree with you that, there are polarization and centralization (kind of siphon effect) processes in cities along HSR lines. As a matter of fact, it is debatable whether small cities can benefit from HSR.

Our previous inappropriate writing is mainly based on the description of the purpose of the establishment of HSR in "China's High-Speed Railway Development." The relevant content is as follows:

Now, china is at a turning point in its urbanization. Small and medium-sized centers—often with only a single main industry, restricting the development of the city—have experienced a continuous loss of talent to large centers. Large cities, by contrast, have high population densities, lack of space, and environmental problems, all affecting business efficiency. HSR aims to reverse this situation by providing a transport backbone that will stimulate complementarity between its cities and allow talent and technology to be devolved to smaller centers to improve their overall competitiveness. These cities will seek to develop their secondary and service industries, and their competitiveness will be strongly influenced by the quality of their transport connections to national and international supply chains and innovation networks.

It now seems unfortunate that we misunderstood a "hope" into reality. However, according to our research, the necessary reason why people in small and medium-sized cities compete for HSR stations (social risk) is that, they believe themselves, or their cities can benefit from HSR. Of course, their perceptions do not necessarily correspond to reality.

Based on this situation, we revised this part carefully in Line 38-40. “The HSR system aims to enhance China’s urbanization process by both encouraging a complementary relationship among cities and allowing talent and technology to flow to smaller centers, thereby enhancing their competitiveness.” Instead of a confirmed fact, we describe the impact of HSR on smaller centers as an opportunity for development.

We try our best to make the description acceptable and not to mislead our readers.

Point 3: Conclusions/results

Please mention the problems of transferability of this methodology (Bayesian network) and approach (social risks of projects) other countries different from China

Response 3: As you suggested that the conclusion should focus more on the transferability, we really agree with your opinion. In Line 551-553, we supplement that the research process and methodology can be applied to the analysis of social risk in other countries, providing profitable reference.

We tried our best to improve the manuscript and made some other changes in the manuscript.

We appreciate for Editors/Reviewer’s warm work earnestly, and hope that the correction will meet with approval.

Once again, thank you very much for your comments and suggestions.

With best regards,

Sincerely yours,

Pengcheng Xiang

Dr. & Prof.

School of Management Science and Real Estate

Chongqing University

Chongqing P.R.China, 400045

Tel: 86-23-65120848

E-mail: [email protected], [email protected].

Round 2

Reviewer 1 Report

The authors made sincere corrections to the review results. And this significantly improved the quality of the overall paper. I think this paper is entitled to publication. However, there are some typos or grammatical errors, so please check carefully and correct them.

Author Response

Response to Reviewer Comment

Dear Editors and Reviewer:

Thank you very much for the editorial staff and reviewer’s hard-work in processing our manuscript (sustainability-724725). We have carefully studied the comments. Also, we have tried our best to revise our manuscript according to the reviewer’s suggestions, and tracked changes in the text for the reviewers’ convenience. We hope this revision can make our paper more acceptable. The comments are addressed below in a point-by-point manner.

Point 1: There are some typos or grammatical errors, so please check carefully and correct them.

Response 1:

Thank you very much for your valuable and thoughtful suggestion. We have carefully rechecked and improved the written English/grammar. Furthermore, we have used a professional English editing service to further improve paper’s written English/grammar. This manuscript was edited for proper English language, grammar, punctuation, spelling, and overall style by one or more of the highly qualified native English speaking editors. There is a proof after the text.

And, we rechecked our manuscript and corrected some formatting errors. Such as the sensitive value ranking in Table 6 and numbering of in-text citations of 39-42. All these corrections can be seen by track change.

We tried our best to improve the manuscript.

We appreciate for Editors/Reviewer’s warm work earnestly, and hope that the correction will meet with approval.

Once again, thank you very much for your comments and suggestions.

With best regards,

Sincerely yours,

Pengcheng Xiang

Dr. & Prof.

School of Management Science and Real Estate

Chongqing University